# RTK+OSNMA Positioning for Road Applications: An Experimental Performance Analysis in Finland

**José M. Vallet García and M. Zahidul H. Bhuiyan ***

Navigation and Positioning Department, Finnish Geospatial Research Institute, National Land Survey of Finland, FI-02150 Espoo, Finland; jose.vallet@nls.fi
* Correspondence: zahidul.bhuiyan@nls.fi

**Abstract:** We compare the performance of dual-band (GPS L1/L2 and Galileo E1/E5a) real-time kinematic (RTK) positioning in an open sky and urban scenarios in southern Finland using two different authentication schemes: one using only satellites authenticated by Galileo's open service navigation message authentication (OSNMA) service (which at the moment of our tests led to using only authenticated Galileo satellites) and the other with no authentication. The results show the actual trade-off between accuracy and availability vs. authenticity associated with using only OSNMA-authenticated satellites, while the authentication of only Galileo satellites is possible (e.g., a drop of RTK positioning availability from 96.67 to 86.01% in our open sky and from 73.55 to 18.65% in our urban scenarios, respectively), and an upper bound of the potential performance that could be reached in similar experimental conditions had the authentication of GPS satellites been supported (e.g., an overall 14 cm and 10.20 m 95% horizontal accuracy in our open sky and urban scenarios, with below 30, 20 and 10 cm during 97.39, 96.03 and 92.43% of the time in the open sky and 49.12, 45.96 and 39.63% in the urban scenarios, respectively).

**Keywords:** RTK; OSNMA; high accuracy; positioning; road applications

## 1. Introduction

The availability of a highly performing localization system is a key prerequisite for the viability of new forthcoming applications in the road sector [1,2]. In order to achieve the required levels of positioning performance and reliability, it is considered necessary to rely on the fusion of information coming from multiple complementary sensors that fill the gaps of each other [3]. Among these, Global Navigation Satellite Systems (GNSSs) are at the center due to their ability to provide global position almost anywhere outdoors. Naturally, then, the accuracy and reliability requirements on GNSS positioning are also increasing.

Among the existing GNSS-based positioning techniques, the one providing the highest accuracy is still real-time kinematic (RTK), with promised (sub) centimeter-level errors. Due to their technological involvement and associated cost, their usage has typically been restricted to specialized professionals. Nowadays, however, these techniques and related variants (e.g., RTK-PPP) are gaining momentum and are being adopted in other domains, the automobile sector being one of them [4].

With respect to the reliability, some weak points of GNSS positioning are its sensitivity to the presence of obstructions and its vulnerability to interference [5]. In relation to the latter, the increasing occurrence of (especially intentional man-made) interferences has prompted the need for their monitoring, detection and even standardization [6,7] not to mention the push on research and development on better detection and mitigation techniques [8,9]. One of them is the new open service navigation message authentication (OSNMA) from the Galileo constellation. This service, the first of its kind, openly provides the means to authenticate the navigation messages and the identity of the satellites used in the position, velocity and time (PVT) computation in a GNSS receiver (henceforth, we

will refer to a satellite as authenticated meaning: that both its identity and transmitted navigation message have been authenticated following the OSNMA protocol). This capability adds an effective layer against attacks using counterfeit signals, also known as spoofing [10], and opens the door to *authenticated positioning*.

Since its conception and proposition, Galileo's OSNMA has undergone a testing, analysis and refinement process [11–14], including a seven-month duration internal preparation phase started on November 2020 [15] and a public observation phase since November 2021. In the public observation phase, interested parties were invited to implement the service following the interface control documents (ICDs) and receiver guidelines [16–19], and to perform tests and report the observed performance. Since then, several open source implementations of the OSNMA protocol have emerged [20–23], and numerous scientific studies have been published discussing its implementation and analyzing its overall performance [13,15,24–34].

While the research still continues toward a deeper analysis and further improvements, a number of initiatives are already focusing on future applications requiring and/or benefiting from authenticated positioning. One example can be found in the Horizon Europe-funded project ESRIUM, under which the investigations pertaining to the present article have been carried out. The project aims at creating highly accurate and authenticated road damage and wear maps for a more efficient maintenance and an increased safety, including the transmission of routing recommendations for manned and/or autonomous vehicles via cooperative intelligent transport systems and services (C-ITS) infrastructure to avoid dangerous areas and even the road wear [35]. Another recent example is the ongoing (also Horizon Europe-funded) CERTIFLIGHT project (Certified European GNSS (E-GNSS) remote tracking of drone and aircraft flights), which aims to utilize OSNMA to certify the flight tracks of drones and ultralight aircraft inside very low level (VLL) airspace [36]. A third example is the European Union project named Drone-borne Galileo Receiver (DEGREE), which is building an OSNMA-enabled receiver for specific category unmanned aerial systems (UAS) operations [37].

A strict requirement on position authenticity as per OSNMA, however, has a price. Firstly, at the moment of this writing ,OSNMA is still in its public observation test phase (since 2021) and can authenticate only Galileo satellites [18], of which there are between 22 and 24 active healthy ones. Therefore, strictly authenticated PVTs are to be computed not only discarding other constellations but also non-authenticated Galileo satellites. Secondly, not all Galileo satellites transmit OSNMA data, and their authenticity is to be assessed using OSNMA data transmitted by other satellites. This scheme is known as satellite *cross-authentication*. The advantage is that less bandwidth is required to upload OSNMA data to the satellites. The drawback is that when a satellite that transmits OSNMA data is not visible to the receiver, the number of authenticated satellites available for an authenticated PVT computation decreases by more than one. This can be an issue especially in obstructed environments such as urban scenarios, where, as shown in [34], OSNMA key performance indicators (KPIs) such as the average number of authenticated satellites, the percentage of authenticated fixes and the time to first authenticated fix (TTFAF) are directly affected by the degree of obstructions experienced by the receiver. All in all, this limited satellite availability results in a decreased positioning performance with respect to what could otherwise be achieved. In the future, however, OSNMA might also authenticate GPS satellites [15], and GPS is also expected to deploy its own signal authentication scheme [38]. In this context, relevant questions are what is the performance that OSNMA-authenticated, RTK-based positioning offers now and what could be the one in that future.

In this article, we present an analysis that tries to quantitatively answer these questions in terms of the overall achieved accuracy and availability in two example scenarios in road applications. The analysis presented here corresponds to data collected in two different environments in southern Finland, namely (a) open sky conditions, essentially with a very low degree of obstructions, and (b) urban, characterized by the presence of buildings that decrease the satellite visibility and create multipaths.

## 2. Operational Scenarios

In this section, we present the details pertaining to the environments and equipment in which the data analyzed in the present study were collected.

### 2.1. Environments

The data collection campaign took place on 4 July 2022, roughly between 13:00 and 17:00 GPS time. Figure 1 shows a map with the trajectory followed during the road tests. The map shows a violet, dashed vertical line dividing the trajectory in two parts. The part of the right side corresponds to urban area (inside Helsinki city), and the one in the left corresponds to open sky (in essence, countryside road and highway). A total of about 400 km were driven, lasting for about four hours. Three of them were spent in the open sky, and one was spent in the urban environments. The speeds varied from 80 to 110 and 0 to 50 km/h in the open and urban environments, respectively.

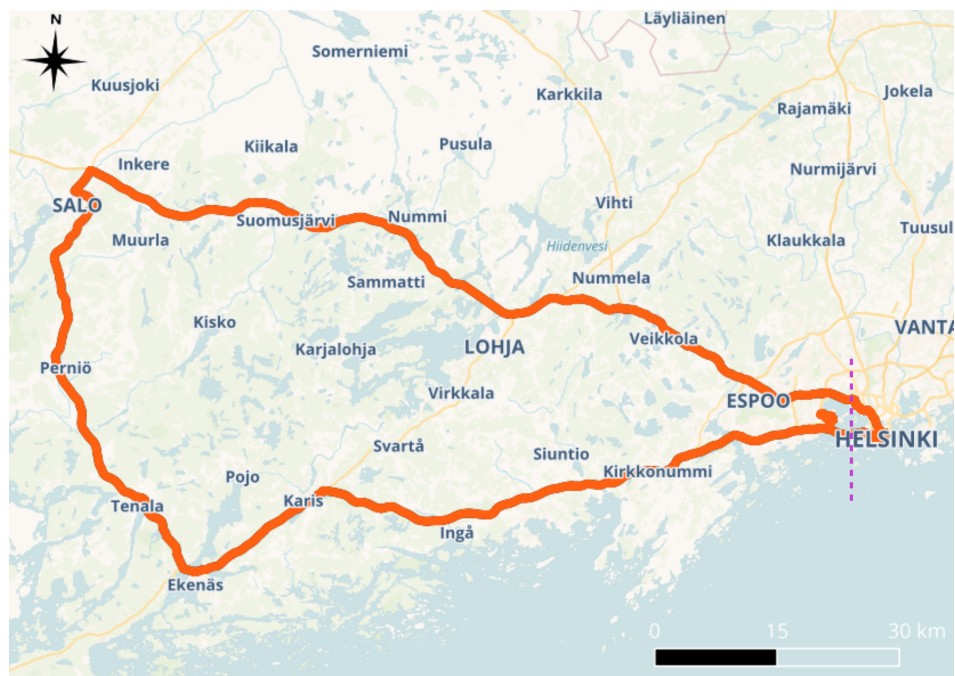

**Figure 1.** Map with the data collection trajectory in red. The violet dashed vertical line divides the trajectory in two areas corresponding to the open-sky (left) and urban (right) environments.

On the highway, there were about 70 bridges like the one presented in Figure 2. This type of infrastructure systematically had an effect on the position estimate, resulting in the outage of RTK-fix positioning during a relatively short time interval. An example of this effect can be seen in the figure, where the receiver fell back to single-point positioning (SPP) after the bridge and then resumed RTK-float first and fix later (as will be seen shortly, the receivers were configured to produce either RTK or SPP positions). The loss of RTK-fix positions was rather systematic over the bridges encountered, and it is expected to occur with similar type of elevated infrastructures.

In addition to bridges, during our tests on the highway we encountered nine tunnels of considerable length which completely blocked the signals from the satellites for a considerable time interval. Figure 3 shows an example, with five of them in a row rather close to each other. Because their most relevant impact is a complete signal blockage, they mostly affect the availability-related KPIs. In this article, the effect of this type of tunnel has been removed by not considering the epochs during which their effect was visible, in particular those from right before entering the tunnels until RTK-fixes were back available after exiting them. Therefore, their presence did not affect the KPIs that will be presented in this article.

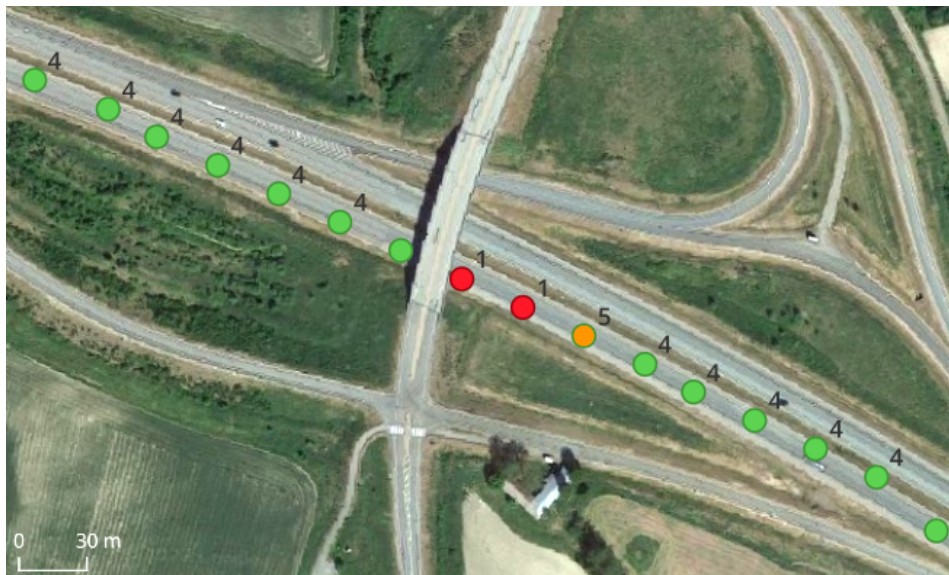

**Figure 2.** Example of the effect of bridges over the road. The car trajectory is marked with colored and labeled dots indicating the solution type in each epoch, as follows: SPP (red, 1), RTK-float (orange, 5) and RTK-fix (green, 4).

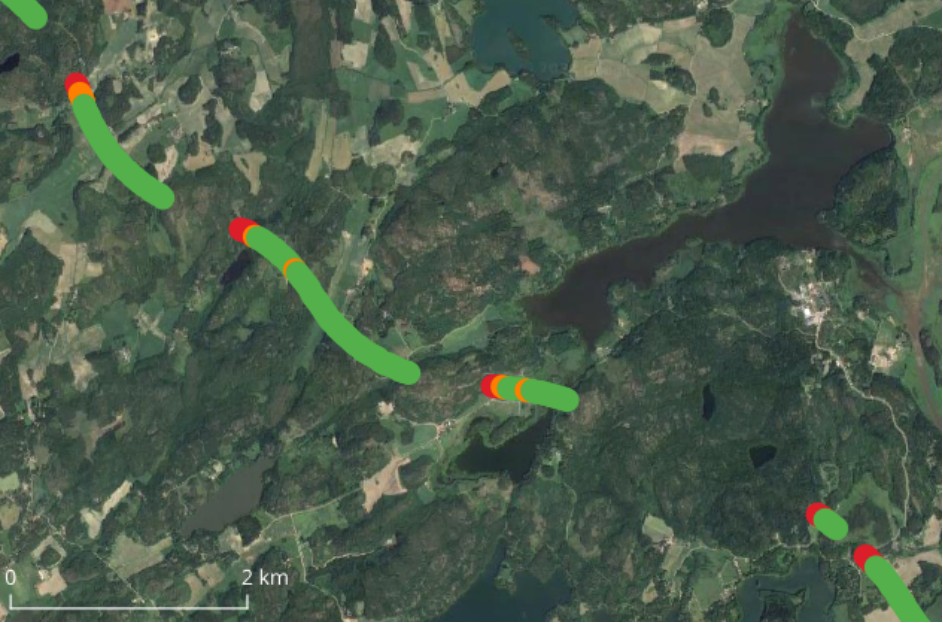

**Figure 3.** Example of tunnel types encountered in the highway. The figure shows five in a row (discontinuities of the dots). The red dots (SPP solutions) mark the exit of the tunnels.

In the urban scenario, the car followed a trajectory so that the buildings' height and street width gradually increased until reaching an area with a notable satellite visibility reduction, which is the area in which most of the measurements were taken. A map of this area and the trajectory followed can be seen in Figure 6 together with the solution types attained when OSNMA was in either of the modes considered.

The time intervals during which the car traversed the open sky and urban scenarios were manually determined by looking at the car trajectory in a map with a satellite view of the driven areas. Broadly speaking, and according to the classification of GNSS reference environments in [39], our highway scenario could be quantitatively classified as predominantly *Flat Rural* (rural clear-sky countryside roads and highways with line of sight (LoS) propagation and masking angles less than 10°) with some parts of *Tree-lined Rural* (rural roads with strong foliage attenuation effects, LoS propagation, and masking angles smaller

than 10°). Our urban scenario would correspond to *European Urban* (narrow streets, large avenues or ring roads in European old big cities with building height from medium to tall, masking angles up to 60° causing frequent non line-of-sight (NLoS) phenomena).

The previous description and classification of the test environments are to be taken as orientative. The intention in this publication is to show a real example of the impact of OSNMA in RTK-based, high-accuracy positioning in environments with different obstruction degrees and not to devise globally applicable quantitative performance metrics. For this purpose, the adopted classification methodology shall suffice. One should not forget that in practice, there are several factors that make the operational scenarios particular to the point that the measured performance can be different even when the tests carried out are thought to be similar. Examples include the following: (a) the satellite geometry and atmospheric conditions (ionosphere and troposphere), which can exhibit particularities depending, e.g., on the receiver position in the globe; (b) the quality of the positioning service provided, which itself depends on the type of RTK service used (e.g., use of the nearest vs. virtual reference stations, etc.), and the quality and density of the reference stations of the service provider's network; and (c) ultimately the local characteristics of the environment and the duration of the tests. It is worth noting in this respect that we have carried out similar tests in Graz, Austria, where the three previously mentioned factors are different. The results from the test carried out in Austria will be published in a forthcoming article.

### 2.2. Equipment

Two Septentrio Mosaic X5 receivers were configured for dual-band (GPS L1/L2 and Galileo E1/E5a) RTK positioning, one with OSNMA off and another with OSNMA in strict mode. The receivers then computed so-called *RTK-fix* or *RTK-float* position estimates when they could and could not solve for the cycle integer ambiguities, respectively [40]. In epochs when RTK was not possible, the receivers were configured to fall back to SPP, where they computed PVT solutions using only the observables that they measured without relying on external assistance/service. The receivers were also configured to compensate for the antenna effects and to use an automotive motion model with moderate acceleration. The position estimates were generated once per second.

Figure 4 schematically shows the interconnection of the devices used to collect the data. Three GNSS receivers were connected to the same geodetic-grade, choke ring antenna (Novatel GPS-703-GGG-HV) via an amplifier/splitter, namely:

- A Propak6 receiver, which was used to collect the GNSS observables together with the readings from a tactical grade inertial measurement unit (IMU) for the computation of the ground truth.
- A Septentrio Mosaic X5 receiver with OSNMA configured in strict mode. This receiver was part of an Ardusimple board with a 4G Networked Transport of RTCM via Internet Protocol (NTRIP) module that was directly retrieving the Radio Technical Comission for Maritime Services (RTCM) messages for RTK positioning from FINPOS, the state owned Finnish positioning services provider [41]. The position of this receiver was recorded in a (control) laptop.
- Another X5 receiver with OSNMA off. This receiver was mounted in a development kit from Septentrio, which itself was connected to the control laptop via USB. The RTCM messages were retrieved using the X5's NTRIP client connecting via the control laptop that was connected to the Internet using a mobile phone (represented by an antenna in the figure).

The RTCM messages used for RTK positioning were obtained from the closest reference station as calculated by FINPOS upon receiving National Marine Electronics Association (NMEA) GGA messages transmitted by the receivers in real time with their approximate position. In total, three different stations were used.

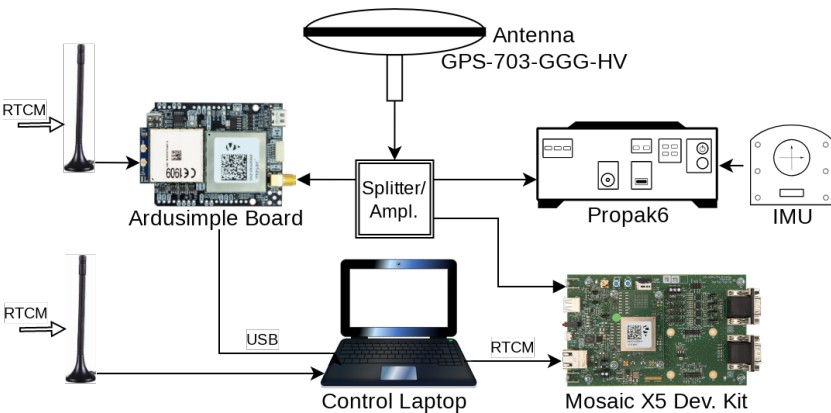

**Figure 4.** Shematic representation of the HW setup.

## 3. Methods

As explained before, the receivers provided both RTK and SPP position types. SPP being a stand-alone technique (i.e., the receiver does not use external assistance such as corrections and/or observables), SPP positions are expressed in the WGS84 frame (the one used to reference the position of the satellites), to which ITRF2014 is considered to agree at the cm level. In the case of RTK positioning, since RTK is a differential technique, they are expressed in the reference frame used to specify the location of the reference stations where the aiding measurements originate. This (so-called *local* or *regional*) frame is realized to optimally represent a specific area/region/country. In Finland, the official local reference frame is EUREF-FIN, which is a realization of ETRS89. ETRS89 is fixed to the Eurasian tectonic plate, whereas WGS84/ITRF2014 are not. Therefore, the coordinates of the same point in space in WGS84/ITRF2014 change as time passes by about 2.5 cm/year due to continental drift. In addition, the Fennoscandian region and its surroundings experience intraplate crustal motions due to the Glacial Isostatic Adjustment (GIA) phenomena (a few mm and up to a cm per year in horizontal and vertical coordinates, respectively). These effects should be considered in high-accuracy positioning by appropriately transforming the coordinates taking them into account. The Nordic Geodetic Commission (NKG) has developed the NKG coordinate transformation methodology that considers both the rigid Eurasian plate motion and the intraplate deformations [42]. In this study, we have used the NKG2020 transformation, which is the latest version of this methodology at the time of this writing and enables an accuracy of a few millimeters [43].

For the computation of errors, the RTK positions were then transformed from EUREF-FIN to ITRF2014 (which agrees with WGS84 at a cm level), properly taking into account the coordinate variations in time due to the previously mentioned deformations. As a result, the coordinates of both RTK and SPP position types were accurately represented in the same coordinate frame (ITRF2014). The errors were then computed with all the coordinates being expressed in this frame.

The ground truth was computed with professional software (SW) (Inertial Explorer) processing tightly coupled measurements from a Novatel SPAN system composed by a Propak6 receiver and a Honeywell HG1700-AG58 tactical grade IMU as well as observables from four FINPOS reference stations distributed along the whole trajectory.

The errors and their statistics presented in this article were computed as follows. Let $x$ be the true position at epoch time $t$ and $\hat{x}$ be a corresponding estimate (a random variable), where the time index is omitted in both cases for brevity. In order to compute the errors at each epoch, we calculate the topocentric projection of $\hat{x}$ centered in $x$. The result is a vector $\delta_{enu} = (\delta_e, \delta_n, \delta_u)$ with the east, north and up components of the projection (henceforth the east–north–up (ENU) error decomposition). We approximate the positioning error as $\hat{x} - x \approx \delta_{enu}$. We will also work with an equivalent longitudinal–transversal–up (LTU) error decomposition $\delta_{ltu} = (\delta_l, \delta_t, \delta_u)$, where the longitudinal and transversal er-

rors $(\delta_l, \delta_t)$ are computed by projecting the horizontal error vector $(\delta_e, \delta_n)$ in the direction along, and perpendicular to, the horizontal component (east and north) of the car velocity, respectively.

We then calculate distance errors in the horizontal and vertical dimensions as $d_h = \sqrt{\delta_e^2 + \delta_n^2} = \sqrt{\delta_l^2 + \delta_t^2}$ and $d_v = \sqrt{\delta_u^2}$, respectively. After computing these for all epochs, we have two sets of distance errors (i.e., positive definite random variables) which can be thought as coming from two random distributions. The accuracy is, in essence, a measure of the information about the error present in these and any other (distance) error distributions. This information can be conveyed in terms of different percentiles (see, e.g., [44,45]). In order to better account for the possible spread of the distributions (which can be heavy-tailed, e.g., in urban environments, see, e.g., [46]); classically, the 95% percentile is used (see, e.g., [40,45]). We then denote Pctl95($d_e$) as the 95% percentile of these distance errors. A high accuracy will then correspond to a low value of this percentile and vice versa. Henceforth, we might loosely use the term *error* to denote, in addition to the epoch-wise error, the 95% percentile of the involved distance error. The difference will be clear from the context.

In addition to statistics of distance errors, we report component-wise biases. Calling $E[.]$ the expectation operator over time, the bias of $\hat{x}$ expressed in its three components is $\boldsymbol{Bias}[\hat{x}] = \boldsymbol{E}[\hat{x}] - \boldsymbol{x} \approx \boldsymbol{E}[\boldsymbol{\delta}]$, which can be expressed using the ENU and LTU decompositions as $\boldsymbol{E}[\boldsymbol{\delta_{enu}}] = \boldsymbol{E}[(\delta_e, \delta_n, \delta_u)] = (b_e, b_n, b_u)$ and $\boldsymbol{E}[\boldsymbol{\delta_{ltu}}] = \boldsymbol{E}[(\delta_l, \delta_t, \delta_u)] = (b_e, b_n, b_u)$, where the expectation is individually computed for each component and is approximated by the (sample) mean. The horizontal and vertical biases are $b_h = \sqrt{b_e^2 + b_n^2} = \sqrt{b_l^2 + b_t^2}$ and $b_v = b_u$, respectively.

We also report availabilities. This metric can be defined in different ways. An important source can be found in a formal framework aiming at defining performance requirements for automotive applications inspired by the concept of performance-based navigation (PBN) from the civil aviation domain (see e.g., [47,48]). In addition to accuracy and availability, this framework defines other performance metrics such as integrity and continuity, which are measures of the trust on the correctness of the provided position information and of the positioning system's ability to provide the required performance during the whole duration of an operation, respectively. These performance indicators are all of special importance in safety and liability critical applications. In this framework, the availability reflects the capability of the positioning system to provide position estimates not only with the required accuracy but also with the required integrity and continuity. In this article, however, we define availability as the proportion of epochs during which a PVT solution is produced by the receiver and its error is smaller than a specified limit, all with respect to the total expected number of position estimates (i.e., if the receiver is configured to produce PVT solutions at a 1 Hz rate, it is expected to produce 3600 solutions in one hour).

The literature in relation to performance requirements is varied [47,49–51], and there is still no widely adopted standard. In this paper, we present horizontal availabilities of positions with errors below 30, 20 and 10 cm simply to give an overall impression of what was achieved in the range of stringent requirements. It is worth noting, however, that the KPIs presented here correspond to GNSS-alone positioning, while, as argued, in a real scenario, other complementary sensors will be used.

## 4. Results

We now proceed to present and comment the KPIs obtained from our analysis. The results are presented aiming to emphasize the impact of the environments and the use of OSNMA on the capability of the receivers to deliver accurate position estimates.

Table 1 presents the percentage of solution types attained (including no solution) with respect to the total number of expected solutions had the receivers produced one in every epoch. The same values are graphically presented in Figure 5 in a stacked bar chart. The main aspects to notice and retain from the presented figure and table follow:

- When not using OSNMA, changing from open sky to urban environments significantly decreased the proportion of RTK-fix positions (which roughly halved) in favor of RTK-float and SPP types.
- Using OSNMA in strict mode significantly decreased the availability of RTK-fix solutions with respect to not using it in both environments (from 96.04 to 68.70% in open sky and from 49.71 to 0.05% in urban).

The effect of this switch in the Helsinki center can be seen in Figure 6, where also the drop of positioning availability in general is clearly visible. Broadly and qualitatively speaking, this suggests that using OSNMA posed significant challenges for leveraging the highest accuracy that RTK can offer.

We now proceed with a quantitative analysis of the positioning performance. For that purpose, Table 2 presents a selection of KPIs computed over different groups of solution types identified by the labels in the first column as follows: (a) SPP + RTK: PVTs from epochs with either SPP or RTK (float or fix) solution types. (b) RTK: PVTs from epochs with RTK (float or fix) solution types. And (c) RTK-fix: PVTs from epochs with RTK-fix solution types. The intention behind this grouping and its presentation order is two-fold. First, it aims at emphasizing the (expected) gradual accuracy increase of the different positioning modes as moving down the group of rows (SPP + RTK → RTK → RTK-fix), and second, it will help us better examine the effective contribution of RTK to high-accuracy positioning in the different environments and OSNMA modes.

For each group, the table presents KPIs for all the combinations of environments and OSNMA modes. The KPIs are presented in groups as follows: First, there are the 95% percentile of the (distance) errors in the longitudinal (along track), transversal (cross-track), horizontal and vertical components, and in three dimensions. Second, there are the biases in the longitudinal, transversal, horizontal and vertical components. A low value of the bias is, in general, an indicator of a good calibration. This is especially the case for the longitudinal and transversal error components, which can clearly reveal an error in the computation of the IMU-to-antenna lever arm. As we can see in the table, these biases are in the order of 4 and less than a mm for RTK-fix positions in open sky condition, which are conditions where the presence of calibration errors would more visibly affect the biases in the longitudinal and transversal components. Third, there is the average number of satellites used in the computation of the corresponding PVT solutions. And fourth, there is the availability of positions with less than 30, 20 and 10 cm of horizontal error.

We now focus on the overall positioning performance (SPP + RTK group). In view of the corresponding KPIs, we present the following main findings:

- Turning OSNMA off significantly decreased the horizontal error in open sky and urban environments. In open sky, the 95% error percentile dropped from 0.75 to 0.145 m, and in urban, it dropped from 18.472 to 10.197 m. This is consistent with the increased proportion of RTK-fix positions presented in Table 1.
- The availabilities severely decreased when switching from open sky to urban environment (e.g., from 97.388 to 49.117% for an error less than 30 cm with OSNMA off). The use of OSNMA in strict mode further decreased the availability in a rather significant manner (to 15.874% following the previous example).
- The average number of satellites used for the position estimation significantly dropped when using OSNMA in strict mode (it roughly halved). As explained before, the main reason is not using the GPS constellation. However, and as already commented in [34], the reliance of OSNMA on satellite cross-authentication can also reduce, to some extent, the availability of authenticated satellites in obstructed environments (not all the visible satellites can be authenticated).

**Table 1.** Percentages of solution types with respect to the expected total number of PVT solutions.

|  | Open Sky | | Urban | |
| --- | --- | --- | --- | --- |
| **OSNMA Strict** | No Sol.: | 03.89% | No Sol.: | 43.73% |
| | SPP: | 08.17% | SPP: | 37.62% |
| | RTK-float: | 19.24% | RTK-float: | 18.59% |
| | RTK-fix: | 68.70% | RTK-fix: | 00.05% |
| | RTK(float+fix): | 87.94% | RTK(float+fix): | 18.65% |
| **OSNMA Off** | No Sol.: | 00.18% | No Sol.: | 00.03% |
| | SPP: | 01.03% | SPP: | 26.42% |
| | RTK-float: | 02.74% | RTK-float: | 23.84% |
| | RTK-fix: | 96.04% | RTK-fix: | 49.71% |
| | RTK(float+fix): | 98.79% | RTK(float+fix): | 73.55% |

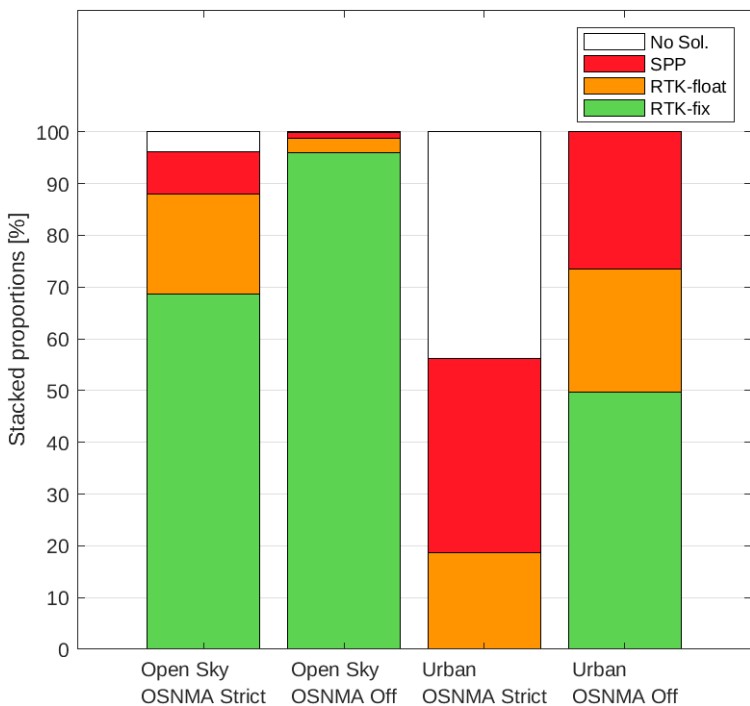

**Figure 5.** Stacked proportion of solution types. The solution types' coloring scheme follows that of Figure 2.

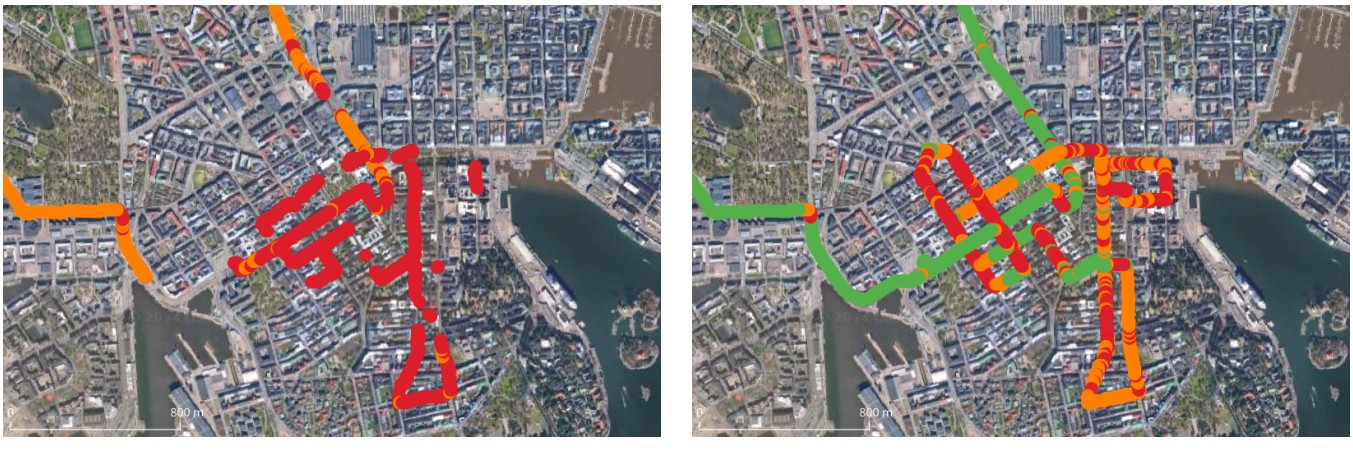

(**a**) OSNMA strict.                    (**b**) OSNMA off.

**Figure 6.** PVT availability and types in Helsinki center. The solution types' coloring scheme follows that of Figure 2.

**Table 2.** KPIs associated to different position types for the different environments and OSNMA modes.

| Sol. Types | Env. | OSNMA Mode | Pctl$_{95}(d_e)$ [m] | | | | | Bias [m] | | | | Av. num. sats. | Avail. (P($d_h$) < $d_e$) [%] | | |
|---|---|---|---|---|---|---|---|---|---|---|---|---|---|---|---|
| | | | Long. | Trans. | Horiz. | Vert. | 3D | Long. | Trans. | Horiz. | Vert. | | $d_e$ = 0.3 | $d_e$ = 0.2 | $d_e$ = 0.1 |
| SPP + RTK | Open | Strict | 0.476 | 0.484 | 0.750 | 0.845 | 1.113 | −0.005 | −0.032 | 0.033 | 0.074 | 5.956 | 84.445 | 80.388 | 74.105 |
| | | Off | 0.088 | 0.097 | 0.145 | 0.259 | 0.332 | −0.004 | −0.002 | 0.005 | 0.057 | 11.197 | 97.388 | 96.026 | 92.429 |
| | Urban | Strict | 6.937 | 17.198 | 18.472 | 25.108 | 31.774 | 0.224 | −1.341 | 1.360 | 4.552 | 4.878 | 15.874 | 7.665 | 4.675 |
| | | Off | 4.380 | 9.588 | 10.197 | 25.711 | 27.362 | 0.356 | 0.492 | 0.608 | 3.882 | 9.678 | 49.117 | 45.964 | 39.630 |
| RTK | Open | Strict | 0.209 | 0.195 | 0.323 | 0.469 | 0.580 | 0.016 | −0.003 | 0.016 | 0.047 | 5.956 | 83.056 | 79.661 | 73.774 |
| | | Off | 0.075 | 0.082 | 0.121 | 0.244 | 0.284 | 0.002 | 0.002 | 0.003 | 0.058 | 11.168 | 97.314 | 95.999 | 92.411 |
| | Urban | Strict | 0.419 | 0.754 | 0.887 | 2.112 | 2.366 | −0.079 | 0.045 | 0.091 | 0.274 | 4.920 | 14.542 | 7.040 | 4.539 |
| | | Off | 1.162 | 1.904 | 2.308 | 3.706 | 4.908 | 0.035 | −0.055 | 0.065 | 1.292 | 8.601 | 48.682 | 45.665 | 39.467 |
| RTK (fix) | Open | Strict | 0.048 | 0.069 | 0.089 | 0.170 | 0.182 | 0.004 | 0.007 | 0.008 | 0.035 | 6.096 | 68.218 | 67.988 | 66.038 |
| | | Off | 0.059 | 0.068 | 0.099 | 0.201 | 0.221 | 0.004 | −0.000 | 0.004 | 0.059 | 11.286 | 95.401 | 94.389 | 91.344 |
| | Urban | Strict | 0.029 | 0.020 | 0.035 | 0.104 | 0.109 | −0.019 | 0.006 | 0.020 | 0.085 | 5.500 | 0.054 | 0.054 | 0.054 |
| | | Off | 0.307 | 0.453 | 0.651 | 1.751 | 1.892 | −0.030 | −0.023 | 0.038 | 0.326 | 9.606 | 42.457 | 41.044 | 37.157 |

We now move to investigating the effective contribution of RTK positioning to high accuracy in our experiments. For that matter, we first center our attention in the rows within the RTK group in Table 2, which considers both float and fix types. From Table 1, we can see that turning OSNMA off increased the percentage of RTK positions from 87.94 to 98.79% in open sky and from 18.65 to 73.55% in the urban environments. This suggests that turning OSNMA off would result in an accuracy and availability increase, which is the case in open sky (e.g., a horizontal error reduction from 32.3 to 12.1 cm and an increase of the 30 cm-level horizontal availability from 83.056 to 97.314%). However, in the urban environment, the errors significantly increased (e.g., the horizontal one from 0.887 to 2.308 m), yet the availabilities also increased (e.g., from 14.542 to 48.682% for errors smaller than 30 cm). The reason can be graphically seen in Figure 7, where the horizontal errors for the RTK float and fix types under the different OSNMA configurations are plotted. As we can see, in the urban environment, the errors for RTK-floats reached large values in a significant number of epochs: in fact, up to 35 m (only errors up to 10 m were plotted to better compare the errors in the open sky scenario). When OSNMA was turned on, the percentage of RTK solutions of both types significantly decreased (see also Table 1 and Figure 5), and when RTK-floats were possible, they reached smaller errors.

We now move our attention to the rows of Table 2 for to RTK-fix solutions. As can be seen, in our open sky scenario with OSNMA off, the horizontal error was 9.9 cm. Interestingly, with OSNMA in strict mode, and despite using on average nearly half the amount of satellites, the error decreased by 1 cm. The corresponding errors can be seen in the lower plots of Figure 7. Investigating the causes of this accuracy increase is out of the scope of this publication. We, however, hypothesize that it was the produce of the use of only intrinsically better Galileo signals ([52,53]). However, the proportion of RTK-fix solutions was significantly lower when OSNMA was in strict mode (68.70 vs. 98.79%, see Table 1). Therefore, its usability in practice decreases, as indicated by the corresponding horizontal availabilities (from 95.401 to 68.218% of positions with errors smaller than 30 cm).

The performance in the urban environment exhibits the same characteristics but more dramatically: while the horizontal error was 3.5 cm with OSNMA in strict mode (vs. 65.1 cm with OSNMA off), the receiver could attain position fixes only in 0.05% of the epochs (see Table 1 and the lower right plot of Figure 7), which led to an availability of 0.054%. Despite the error being significantly larger when OSNMA was off, its availability was also significantly higher (42.457%). This essentially means that the highest accuracy that RTK can deliver could not be exploited when OSNMA was in strict mode.

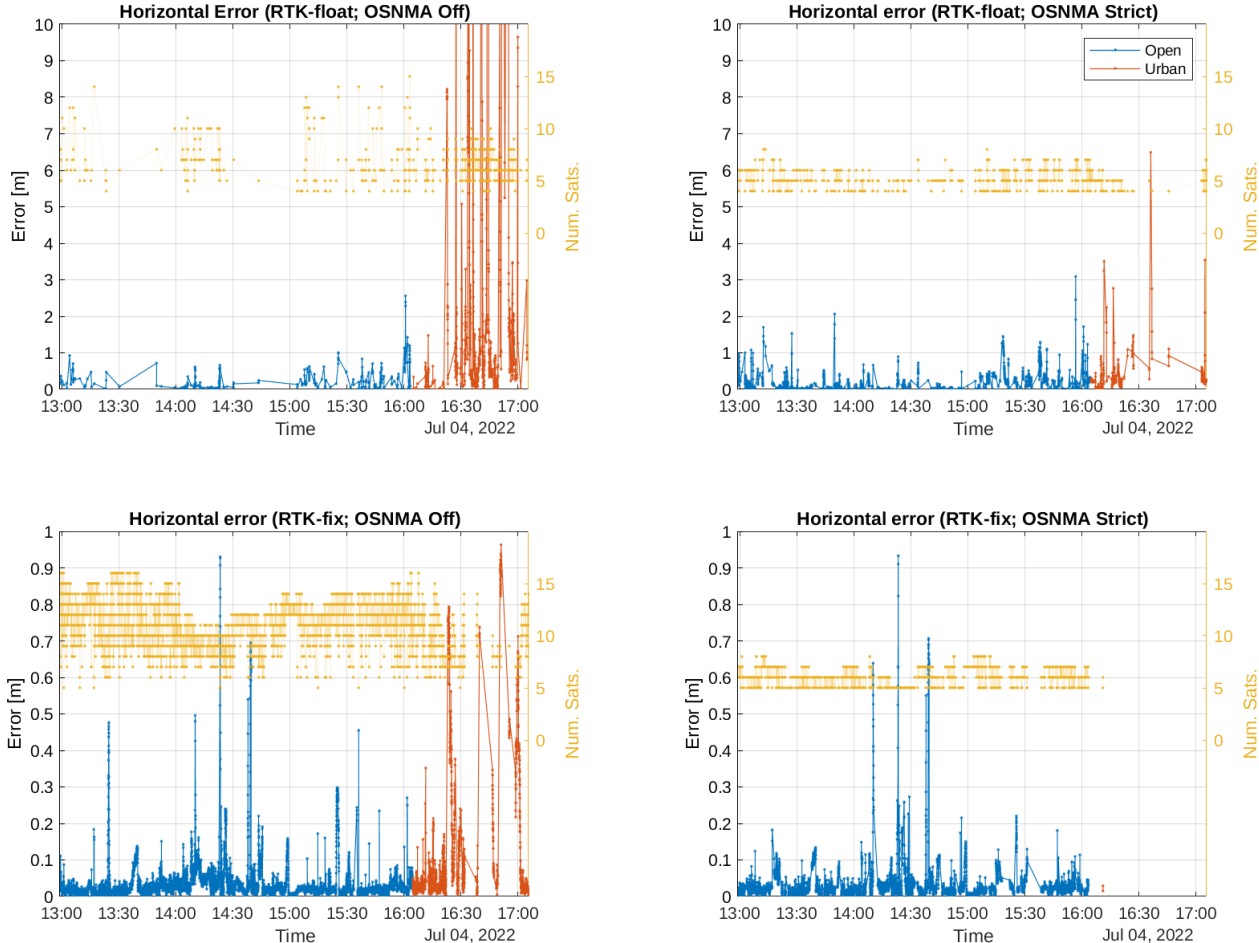

**Figure 7.** Errors (horizontal and vertical) and number of satellites used vs. time for RTK-fix positions. The legends in the upper-right panel are common to all plots.

Finally, Figure 8 presents scatter plots of the horizontal errors associated to RTK fix and float positions, where the overall impact of the environment and OSNMA on the potential high accuracy can be easily appreciated graphically. Comparing for example the left panels from top to bottom (which correspond to switching off OSNMA in the open sky environment), one can clearly see the proportion increase of RTK-fix solutions (or, equivalently, the decrease of RTK float ones) together with the corresponding overall accuracy increase (concentration of the cloud points). We can also see how when OSNMA was in strict mode, the RTK-fix solutions were slightly more accurate (with 0.089 vs. 0.099 m horizontal error, see Table 2), although they were attained in a significantly smaller proportion (68.70 vs. 96.04%; see Table 1), resulting in also a significantly lower horizontal availability (e.g., 95.401 vs. 68.218% for less than 30 cm error, see Table 2). Similarly, when switching off OSNMA in the urban scenario (as shown by moving from top-right to bottom-right panels), we can see a significant proportion increase of RTK-fix solutions, which were in practice unavailable with OSNMA in strict mode (see Table 1). We also again see how despite the overall RTK accuracy being higher when OSNMA was in strict mode (0.887 vs. 2.308 m, see Table 2), this comes at the expense of a significantly lower horizontal availability (e.g., 48.682 vs. 14.542% for less than 30 cm error; see Table 2).

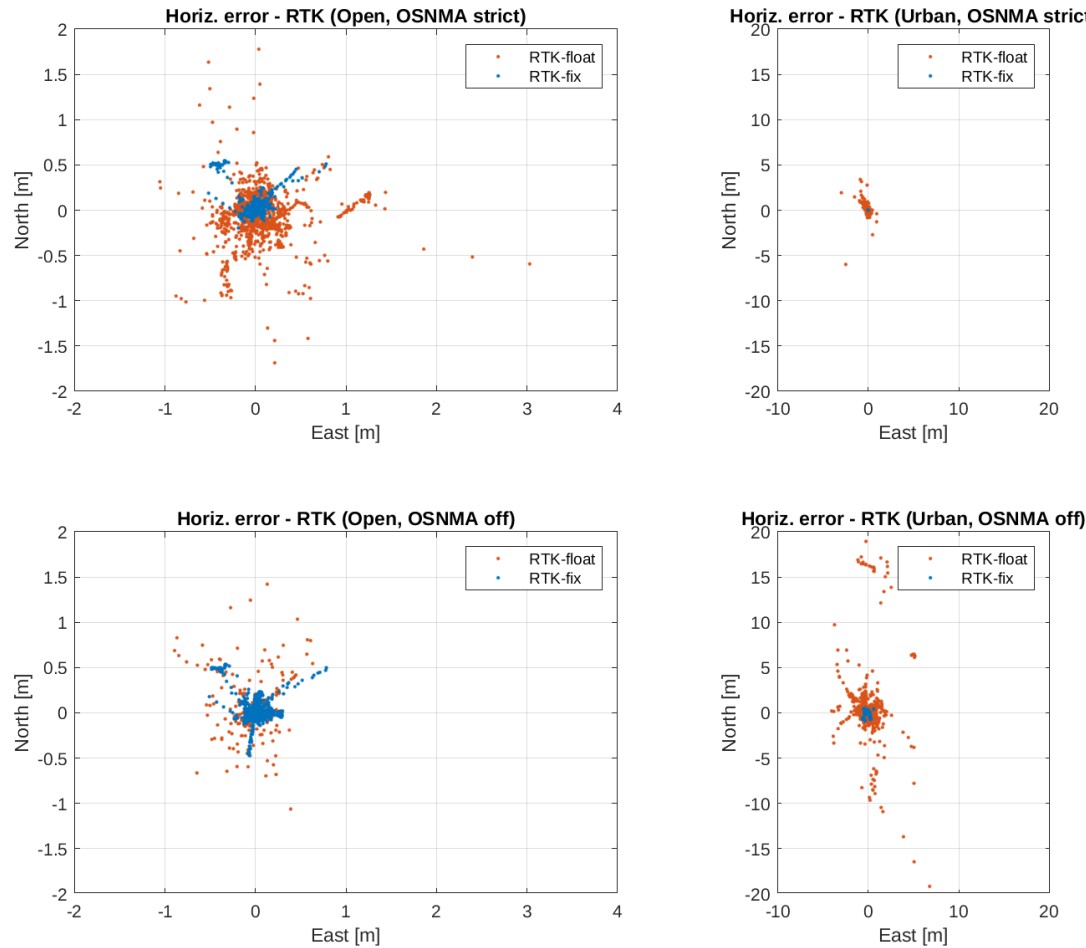

**Figure 8.** Scatter plot of the horizontal errors (east and north components) corresponding to RTK positions in both environments and OSNMA configurations.

## 5. Discussion

The results presented can be used first to evaluate the impact of a strict authentication as per OSNMA in its actual state (in the public testing phase and supporting the authentication of only Galileo satellites) on the overall (high accuracy) positioning performance and second to have a first approximation on what performance could be achieved in similar operational scenarios if the authentication of GPS navigation messages becomes supported. Indeed, broadly speaking and under the assumption that no real spoofing occurred during our tests, we can take the observed performance corresponding to having OSNMA off as an upper bound of what could have been attained in similar environmental conditions and with similar equipment and receiver configuration had OSNMA supported the authentication of GPS navigation messages. The results cannot be, however, taken as fully equivalent due to the reliance of OSNMA on cross-authentication, which entails a non-negligible probability of having visible satellites for which the receiver does not have OSNMA data and, consequently, for which OSNMA cannot be used to ascertain (nor discredit) their authenticity. In these cases, a strict authenticity requirement as per OSNMA would preclude the use of those satellites in the PVT computation. In addition, and in relation to possible performance in the future, it is expected that it will increase as more Galileo satellites are launched and as the OSNMA observation phase advances, especially as more Galileo satellites are configured to send OSNMA data and as other possible improvements are implemented. Nevertheless, and although it is reasonable to expect different quantitative results from tests carried out in different operational scenarios and with improvements over time, there are some trends that are likely to remain.

The results presented in the previous section clearly indicate that the authenticity has a price. First, the percentage of RTK-fix solutions can significantly decrease in favor of (the less performing) RTK-foats and SPPs (see Table 1 and Figure 5). And second, the overall availability can consequently decrease also significantly (see the values for the RTK+SPP group of Table 2). However, the extent and significance of this reduction strongly depends on the environment (by a factor of 5 and 1.8 for the open sky and urban environments, respectively), which in turn percolates the effect of other factors.

The fact that the errors for RTK-fix solutions were smaller when OSNMA was in strict mode (using roughly half the number of satellites than with OSNMA in off mode, and all from Galileo) reveals the special features of the Galileo signals and the potential benefits derived from their usage. It also suggests that increasing the availability of authenticated Galileo satellites would be especially beneficial for increasing the accuracy and availability. In addition to having a larger number of deployed satellites, this can also be accomplished, although to a limited extent, by increasing the number of satellites that transmit OSNMA data (a configurable parameter in OSNMA [18]), which would increase the probability of successful cross-authentications. This would be especially beneficial in urban environments [34], where almost no RTK-fix was attained when OSNMA was in strict mode in our experiments.

In any case, ultimately the accuracy has to be evaluated together with the availability in order to better assess the usability of the considered positioning scheme. This is especially the case when requiring RTK solutions only; as we have seen, they can show appealing accuracies but be only marginally attained.

Regarding the quality of a possible future OSNMA service that also authenticates GPS satellites and navigation messages, the presented results show that the overall horizontal accuracy would have been of (up to) 14.5 cm and 10.197 m in the open sky and urban environments, with below $-30$ cm horizontal errors available 97.388 and 49.117% of the time in the open sky and urban scenarios, respectively. However, it is worth noting that the quantitative results can be heavily dependent on the particularities of the operational scenarios. This has been observed in additional tests carried out in Graz (Austria) within the same project, the results of which will be analyzed in detail in a forthcoming complementary publication.

## 6. Conclusions

We have conducted tests in open sky and urban environments to study the performance of dual band (GPS L1/L2 + Galileo E1/E5a) RTK positioning (with fallback to SPP) authenticated by means of OSNMA in road applications. The results clearly indicate that aiming at a strict authentication (which at the moment imposes using only authenticated Galileo satellites) can result in a significant degradation of the overall performance and usability with respect to not using authentication at all, as indicated by significant increases of the 95% horizontal error percentiles (e.g., from 14.5 to 75 cm in our open sky and from 10.197 to 18.472 m in our urban scenarios, respectively) as well as drops in the availability (e.g., from 97.388 to 84.445% of solutions with an error smaller than 30 cm in our open sky scenario and from 49.117 to 15.874% in our urban environment). We have also seen that the low availability of Galileo satellites holds back the potential behind the special characteristics of its signals. The results on the other hand also suggest that if the authentication of GPS navigation messages becomes a reality, receivers in similar operational scenarios would be able to deliver authenticated PVTs with overall horizontal accuracies of (up to) 14.5 cm and 10.197 m in the open sky and urban environments, which could be available with smaller than 30 cm errors 97.388 and 49.117% of the time, respectively. Although it is reasonable to expect similar trends in similar tests, the particularities of the operational scenario (e.g., geographical area, atmospheric conditions, position provider used, etc.) can visibly impact the quantitative results, and dedicated longer tests are to be conducted when a more plausible and exact performance assessment is needed.

**Author Contributions:** Conceptualization, J.M.V.G.; methodology, J.M.V.G.; software, J.M.V.G.; validation, J.M.V.G. and M.Z.H.B.; formal analysis, J.M.V.G.; investigation, J.M.V.G.; resources, J.M.V.G.; data curation, J.M.V.G.; writing—original draft preparation, J.M.V.G.; writing—review and editing, J.M.V.G. and M.Z.H.B.; visualization, J.M.V.G.; supervision, J.M.V.G.; project administration, J.M.V.G. and M.Z.H.B.; funding acquisition, M.Z.H.B. All authors have read and agreed to the published version of the manuscript.

**Funding:** This work was conducted in the scope of the ESRIUM Project, which has received funding from EUSPA as part of EU-Horizon 2020 research and innovation programme under grant agreement No 101004181. The content of this paper reflects only the authors' view. Neither the European Commission nor the EUSPA is responsible for any use that may be made of the information it contains.

**Institutional Review Board Statement:** Not applicable.

**Informed Consent Statement:** Not applicable.

**Data Availability Statement:** The datasets presented in this article are not readily available because the data are part of an ongoing study. Requests to access the datasets should be directed to the corresponding author.

**Acknowledgments:** The authors would like to thank Pasi Häkli for his assistance with coordinate transformations and associated accuracy matters.

**Conflicts of Interest:** The authors declare no conflicts of interest.

## Abbreviations

The following abbreviations are used in this manuscript:

| | |
|---|---|
| C-ITS | cooperative intelligent transport systems and services |
| E-GNSS | European GNSS |
| ENU | east–north–up |
| GIA | Glacial Isostatic Adjustment |
| GNSS | Global Navigation Satellite System |
| ICD | interface control document |
| IMU | inertial measurement unit |
| KPI | key performance indicator |
| LoS | line of sight |
| LTU | longitudinal–transversal–up |
| NKG | Nordic Geodetic Commission |
| NLoS | non line-of-sight |
| NMEA | National Marine Electronics Association |
| NTRIP | Networked Transport of RTCM via Internet Protocol |
| OSNMA | open service navigation message authentication |
| PBN | performance-based navigation |
| PVT | position, velocity and time |
| RTCM | Radio Technical Comission for Maritime Services |
| RTK | real-time kinematic |
| SPP | single-point positioning |
| SW | software |
| TTFAF | time to first authenticated fix |
| UAS | unmanned aerial systems |
| VLL | very low level |

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
