# Peer review of "RTK+OSNMA Positioning for Road Applications: An Experimental Performance Analysis in Finland"

_sensors, doi:10.3390/s24020621_

Round 1

Reviewer 1 Report

Comments and Suggestions for Authors

1-) Why this issue (authentication) is important should be discussed with the recent literature.

2-) Line 76 reads "Roughly speaking,...". I do not think this is an appropriate phrase for an academic article.

3-) Line 163: the parenthesis in brackets and the two e.g. statements should be corrected (e.g., the receiver does not use external assistance (e.g. corrections and/or observables),

4-) Especially in "OSNMA Strict" mode, the accuracy in meters will not be sufficient for a road project that mostly requires accuracy in cm-dm. If this mode is activated, GNSS positioning will be almost impossible, especially in the urban environment. Do you think this is not a problem?

5-) Since OSNMA is only for Galileo satellites, the comparison with GPS will only be possible in "OSNMA-off" mode. Therefore, it would be more meaningful to compare "OSNMA Strict" with "OSNMA-off" only with Galileo-only solutions.

In this context, when comparing the results, it seems that it would be more meaningful to compare "OSNMA Strict with Galileo-only" and "OSNMA-off with Galileo-only" with GPS-only in order to demonstrate the effect of OSNMA authentication on the positioing performance. In this way, I think it will be better understood why I should (or should not) use OSNMA.

Author Response

Dear Reviewer,

Thank you for your constructive review. We answered and modified relevant part of the submitted manuscript based on your comments. 

Best Wishes

Prof. Zahidul Bhuiyan

Reviewer 2 Report

Comments and Suggestions for Authors

1.     Has the conclusion that RTK-fix positions are systematically lost at all bridges been validated? Has this conclusion been validated for similar elevated infrastructure? Has it been considered what the pattern of change in RTK-fix positions would be if a slightly more complex situation, such as a circular viaduct, were to be encountered, creating multiple occlusions?

2.     A simple threshold has been given for considering availabilities; have non-empirical thresholds been considered? Is it possible to refer to the concept of integrities? Is it possible to refer to the notion of integrity rate of observations, not depending on the frequency of solutions.

3.     What does ’..., where the presence of 268 these errors would more visibly affect these biases.’ mean in the first paragraph of page 9 of the text?

4.     The position of the data reference is incorrect, in the second to last line of the last paragraph on page 10, it is "(0.887 vs. 2.308 m, see 1)".

5.     Writing error or data error. The value of "48.682 vs. 15.542%" in the last paragraph of page 10 is inconsistent with the value in Table 2.

6.     Is it possible to achieve this effect in any satellite positioning system by adding OSNMA verification services, and the more available satellites, the more obvious the effect.

7.     The article did not mention many topics related to IMU and multipath. Is it abrupt when it appears in the part of Discussion.

Author Response

(The authors gave the same response as above.)

Round 2

Reviewer 1 Report

Comments and Suggestions for Authors

The authors have made the necessary corrections (or provided satisfactory explanations) regarding the issues mentioned by me. The article can be accepted in its version. I congratulate the authors who study this interesting subject.